# REASONING-MODULATED REPRESENTATIONS

## ABSTRACT

Neural networks leverage robust internal representations in order to generalise. Learning them is difficult, and often requires a large training set that covers the data distribution densely. We study a common setting where our task is not purely opaque. Indeed, very often we may have access to information about the underlying system (e.g. that observations must obey certain laws of physics) that any "tabula rasa" neural network would need to re-learn from scratch, penalising data efficiency. We incorporate this information into a pre-trained reasoning module, and investigate its role in shaping the discovered representations in diverse self-supervised learning settings from pixels. Our approach paves the way for a new class of data-efficient representation learning.

## 1 INTRODUCTION

Neural networks are able to learn policies in environments without access to their specifics (Schrittwieser et al., 2020), generate large quantities of text (Brown et al., 2020), or automatically fold proteins to high accuracy (Senior et al., 2020). However, such "tabula rasa" approaches hinge on having access to substantial quantities of data, from which robust representations can be learned. Without a large training set that spans the data distribution, representation learning is difficult (Belkin et al., 2019; LeCun, 2018; Sun et al., 2017).

Here, we study ways to construct neural networks with representations that are robust, while retaining a data-driven approach. We rely on a simple observation: very often, we have some (partial) knowledge of the underlying dynamics of the data, which could help make stronger predictions from fewer observations. This knowledge, however, usually requires us to be mindful of *abstract* properties of the data—and such properties cannot always be robustly extracted from *natural* observations.

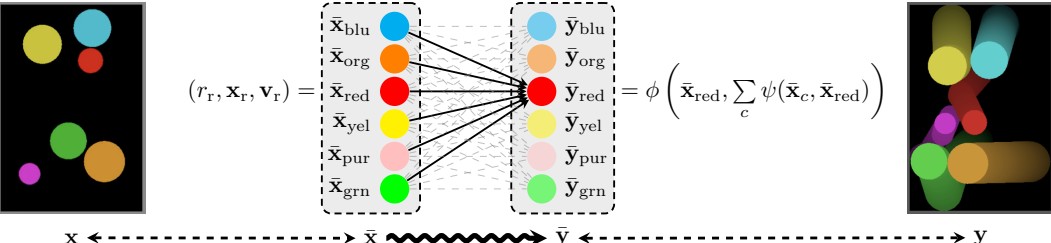

Figure 1: Bouncing balls example (re-printed, with permission, from Battaglia et al. (2016)). Natural inputs, $\mathbf{x}$, correspond to pixel observations. Predicting future observations (natural outputs, $\mathbf{y}$), can be simplified as follows: if we are able to extract a set of *abstract* inputs, $\bar{\mathbf{x}}$, (e.g. the radius, position and velocity for each ball), the movements in this space must obey the laws of physics.

**Motivation** Consider the task of predicting the future state of a system of $n$ bouncing balls, from a pixel input $\mathbf{x}$ (Figure 1). Reliably estimating future pixel observations, $\mathbf{y}$, is a challenging reconstruction task. However, the generative properties of this system are simple. Assuming knowledge of simple abstract inputs (radius, $r_c$, position, $\mathbf{x}_c$, and velocity, $\mathbf{v}_c$) for every ball, $\bar{\mathbf{x}}_c$, the future movements in this abstract space are the result of applying the laws of physics to these low-dimensional quantities. Hence, future abstract states, $\bar{\mathbf{y}}$, can be computed via a simple algorithm that aggregates pair-wise forces between objects.

While this gives us a potentially simpler path from pixel inputs to pixel outputs, via abstract inputs to abstract outputs ($\mathbf{x} \to \bar{\mathbf{x}} \rightsquigarrow \bar{\mathbf{y}} \to \mathbf{y}$), it still places potentially unrealistic demands on our task setup, every step of the way:

$\mathbf{x} \to \bar{\mathbf{x}}$: Necessitates either upfront knowledge of how to abstract away $\bar{\mathbf{x}}$ from $\mathbf{x}$, or a massive dataset of *paired* $(\mathbf{x}, \bar{\mathbf{x}})$ to learn such a mapping from;

$\bar{\mathbf{x}} \rightsquigarrow \bar{\mathbf{y}}$: Implies that the algorithm *perfectly* simulates all aspects of the output. In reality, an algorithm may often only give partial context about $\mathbf{y}$. Further, algorithms often assume that $\mathbf{x}$ is provided without error, exposing an algorithmic bottleneck (Deac et al., 2021): if $\bar{\mathbf{x}}$ is incorrectly predicted, this will negatively compound in $\bar{\mathbf{y}}$, hence $\mathbf{y}$;

$\bar{\mathbf{y}} \to \mathbf{y}$: Necessitates a renderer that generates $\mathbf{y}$ from $\bar{\mathbf{y}}$, or a dataset of *paired* $(\bar{\mathbf{y}}, \mathbf{y})$ to learn it.

We will assume a general setting where *none* of the above constraints hold: we know that the mapping $\bar{\mathbf{x}} \rightsquigarrow \bar{\mathbf{y}}$ is likely of use to our predictor, but we do not assume a trivial mapping or a paired dataset which would allow us to convert directly from $\mathbf{x}$ to $\bar{\mathbf{x}}$ or from $\bar{\mathbf{y}}$ to $\mathbf{y}$. Our only remaining assumption is that the algorithm $\bar{\mathbf{x}} \rightsquigarrow \bar{\mathbf{y}}$ can be efficiently computed, allowing us to generate massive quantities of paired abstract input-output pairs, $(\bar{\mathbf{x}}, \bar{\mathbf{y}})$.

**Present work**  In this setting, we propose Reasoning-Modulated Representations (RMR), an approach that first learns a latent-space *processor* of abstract data; i.e. a mapping $\bar{\mathbf{x}} \xrightarrow{f} \mathbf{z} \xrightarrow{P} \mathbf{z}' \xrightarrow{g} \bar{\mathbf{y}}$, where $\mathbf{z} \in \mathbb{R}^k$ are high-dimensional latent vectors. $f$ and $g$ are an encoder and decoder, designed to take abstract representations to and from this latent space, and $P$ is a processor network which simulates the algorithm $\bar{\mathbf{x}} \rightsquigarrow \bar{\mathbf{y}}$ in the latent space.

We then observe, in the spirit of neural algorithmic reasoning (Veličković & Blundell, 2021), that such a processor network can be used as a drop-in differentiable component for *any* task where the $\bar{\mathbf{x}} \rightsquigarrow \bar{\mathbf{y}}$ kind of reasoning may be applicable. Hence, we then learn a pipeline $\mathbf{x} \xrightarrow{\tilde{f}} \mathbf{z} \xrightarrow{P} \mathbf{z}' \xrightarrow{\tilde{g}} \mathbf{y}$, which *modulates* the representations $\mathbf{z}$ obtained from $\mathbf{x}$, forcing them to pass through the pre-trained processor network. By doing so, we have ameliorated the original requirement for a massive *natural* dataset of $(\mathbf{x}, \mathbf{y})$ pairs. Instead, we inject knowledge from a massive *abstract* dataset of $(\bar{\mathbf{x}}, \bar{\mathbf{y}})$ pairs, directly through the pre-trained parameters of $P$. This has the potential to relieve the pressure on encoders and decoders $\tilde{f}$ and $\tilde{g}$, which we experimentally validate on several challenging representation learning domains.

Our contributions can be summarised as follows:

- Verifying and extending prior work, we show that meaningful latent-space models can be learned from massive abstract datasets, on physics simulations and Atari 2600 games;
- We then show that these latent-space models can be used as differentiable components within neural pipelines that process raw observations. In doing so, we recover a data-efficient neural network pipeline that relies solely on the existence of massive abstract datasets (which can often be automatically generated).
- Finally, we demonstrate early signs of processor *reusability*: latent-space abstract models can be used in tasks which do not even directly align with their environment, so long as these tasks can benefit from their underlying reasoning procedure.

## 2  RELATED WORK

**Neural algorithmic reasoning**  RMR relies on being able to construct robust latent-space models that imitate abstract reasoning procedures. This makes it well aligned with neural algorithmic reasoning (Cappart et al., 2021), which is concerned with constructing neuralised versions of classical algorithms (typically by learning to execute them in a manner that extrapolates). Leveraging the ideas of algorithmic alignment (Xu et al., 2019), several known algorithmic primitives have already been successfully neuralised. This includes *iterative computation* (Veličković et al., 2019; Tang et al., 2020), *linearithmic algorithms* (Freivalds et al., 2019), and *data structures* (Strathmann et al., 2021; Veličković et al., 2020). Further, the XLVIN model (Deac et al., 2021) demonstrates how such primitives can be re-used for *data-efficient* planning, paving the way for a blueprint (Veličković & Blundell, 2021) that we leverage in RMR as well.

**Physical simulation with neural networks**   Our work also has contact points with prior art in using (graph) neural networks for *physics simulations*. In fact, there is a tight coupling between algorithmic computation and simulations, as the latter are typically realised using the former. Within this space, abstract GNN models of physics have been proposed by works such as interaction networks (Battaglia et al., 2016) and NPE (Chang et al., 2016), and extended to pixel-based inputs by visual interaction networks (Watters et al., 2017). The generalisation power of these models has increased drastically in recent years, with effective models of systems of particles Sanchez-Gonzalez et al. (2020) as well as meshes (Pfaff et al., 2020) being proposed. Excitingly, it has also been demonstrated that rudimentary laws of physics can occasionally be recovered from the update rules of these GNNs (Cranmer et al., 2020), and that they can be used to uncover new physical knowledge (Bapst et al., 2020).

Recent work has also explored placing additional constraints on learning-based physical simulators, for example by using Hamiltonian ODE integrators in conjunction with GNN models (Sanchez-Gonzalez et al., 2019), or by coupling a (non-neural) differentiable physics engine directly to visual inputs and optimizing its parameters via backprop (Jaques et al., 2019; Jatavallabhula et al., 2021).

**Object-centric and modular models for dynamic environments**   RMR with factored latents can be viewed as a form of *object-centric neural network*, in which visual objects in an image or video are represented as separate latent variables in the model and their temporal dynamics and pairwise interactions are modeled via GNNs or self-attention mechanisms. There is a rich literature on discovering objects and learning their dynamics from raw visual data without supervision, with object-centric models such as R-NEM (Van Steenkiste et al., 2018), SQAIR (Kosiorek et al., 2018), OP3 (Veerapaneni et al., 2020), SCALOR (Jiang et al., 2019), G-SWM (Lin et al., 2020). Recent work has explored using contrastive losses (Kipf et al., 2019) in this context or other losses directly in latent space (François-Lavet et al., 2019). Related approaches discover and use keypoints (Kulkarni et al., 2019) to describe objects and even discover causal relations from visual input (Li et al., 2020) using neural relational inference (Kipf et al., 2018) in conjunction with a keypoint discovery method. A related line of works integrate attention-mechanisms in modular and object-centric models to interface latent variables with visual input, including models such as RMC (Santoro et al., 2018), RIM (Goyal et al., 2019), Slot Attention (Locatello et al., 2020), SCOFF (Goyal et al., 2021b), and NPS (Goyal et al., 2021a).

## 3   RMR ARCHITECTURE

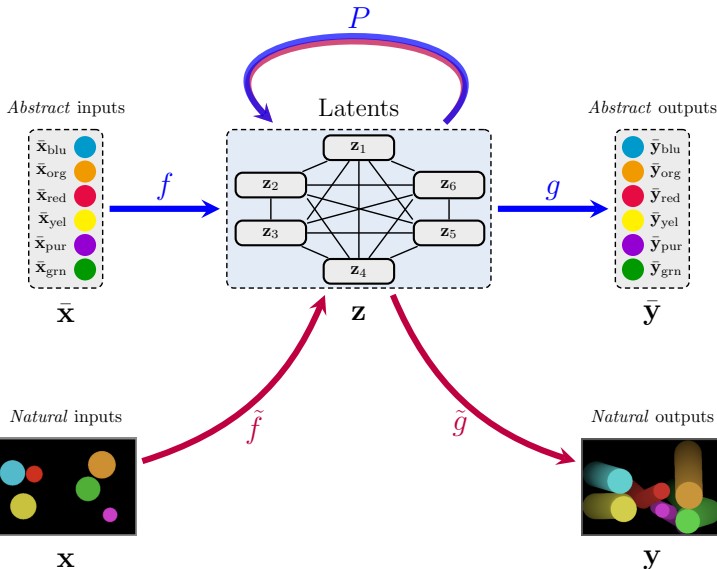

Figure 2: Reasoning-modulated representation learner (RMR).

Having provided a high-level overview of RMR and surveyed the relevant related work, we proceed to carefully detail the blueprint of RMR's various components. This will allow us to ground any subsequent RMR experiments on diverse domains directly in our blueprint. Throughout this section, it will be useful to refer to Figure 2 which presents a visual overview of this section.

**Preliminaries**  We assume a set of *natural inputs*, $\mathcal{X}$, and a set of *natural outputs*, $\mathcal{Y}$. These sets represent the possible inputs and outputs of a target function, $\Phi : \mathcal{X} \to \mathcal{Y}$, which we would like to learn based on a (potentially small) dataset of input-output pairs, $(\mathbf{x}, \mathbf{y})$, where $\mathbf{y} = \Phi(\mathbf{x})$.

We further assume that the inner workings of $\Phi$ can be related to an *algorithm*, $A : \bar{\mathcal{X}} \to \bar{\mathcal{Y}}$. The algorithm operates over a set of *abstract inputs*, $\bar{\mathcal{X}}$, and produces outputs from an *abstract output set* $\bar{\mathcal{Y}}$. Typically, it will be the case that $\dim \bar{\mathcal{X}} \ll \dim \mathcal{X}$; that is, abstract inputs are assumed substantially lower-dimensional than natural inputs. We do not assume existence of any aligned input pairs $(\mathbf{x}, \bar{\mathbf{x}})$, and we do not assume that $A$ perfectly explains the computations of $\Phi$. What we do assume is that $A$ is either known or can be trivially computed, giving rise to a massive dataset of abstract input-output pairs, $(\bar{\mathbf{x}}, \bar{\mathbf{y}})$, where $\bar{\mathbf{y}} = A(\bar{\mathbf{x}})$.

Lastly, we assume a *latent space*, $\mathcal{Z}$, and that we can construct neural network components to both encode and decode from it. Typically, $\mathcal{Z}$ will be a real-valued vector space ($\mathcal{Z} = \mathbb{R}^k$) which is high-dimensional; that is, $k > \dim \bar{\mathcal{X}}$. This ensures that any neural networks operating over $\mathcal{Z}$ are not vulnerable to bottleneck effects.

Note that either the natural or abstract input set may be *factorised*, e.g., into objects; in this case, we can accordingly factorise the latent space, enforcing $\mathcal{Z} = \mathbb{R}^{n \times k}$, where $n$ is the assumed maximal number of objects (typically a hyperparameter of the models if not known upfront).

**Abstract pipeline**  RMR training proceeds by first learning a model of the algorithm $A$, which is bound to pass through a latent-space representation. That is, we learn a neural network approximator $g(P(f(\bar{\mathbf{x}}))) \approx A(\bar{\mathbf{x}})$, which follows the encode-process-decode paradigm (Hamrick et al., 2018). It consists of the following three building blocks: Encoder, $f : \bar{\mathcal{X}} \to \mathcal{Z}$, tasked with projecting the abstract inputs into the latent space; Processor, $P : \mathcal{Z} \to \mathcal{Z}$, simulating individual steps of the algorithm in the latent space; Decoder, $g : \mathcal{Z} \to \bar{\mathcal{Y}}$, tasked with projecting latents back into the abstract output space. Such a pipeline is now widely used both in neural algorithmic reasoning (Veličković et al., 2019) and learning physical simulations (Sanchez-Gonzalez et al., 2020), and can be trained end-to-end with gradient descent.

For reasons that will become apparent, it is favourable for most of the computational effort to be performed by $P$. Accordingly, encoders and decoders are often designed to be simple learnable linear projections, while processors tend to be either deep MLPs or graph neural networks—depending on whether the latent space is factorised into nodes.

**Natural pipeline**  Once an appropriate processor, $P$, has been learned, it may be observed that it corresponds to a highly favourable component in our setting. Namely, we can relate its operations to the algorithm $A$, and since it stays high-dimensional, it is a differentiable component we can easily plug into other neural networks without incurring any bottleneck effects. This insight was originally recovered in XLVIN (Deac et al., 2021), where it yielded a generic implicit planner. As an ablation, we have also rediscovered the bottleneck effect in our settings; see Appendix C. We now leverage similar insights for general representation learning tasks.

On a high level, what we need to do is simple and elegant: swap out $f$ and $g$ for *natural* encoders and decoders, $\tilde{f} : \mathcal{X} \to \mathcal{Z}$ and $\tilde{g} : \mathcal{Z} \to \mathcal{Y}$, respectively. We are then able to learn a function $\tilde{g}(P(\tilde{f}(\mathbf{x}))) \approx \Phi(\mathbf{x})$, which is once again to be optimised through gradient descent. We would like $P$ to retain its semantics during training, and therefore it is typically kept *frozen* in the natural pipeline. Note that $P$ might not perfectly represent $A$ which in turn might not perfectly represent $\Phi$. While we rely on a skip connection in our implementation of $P$, it has no learnable parameters and does not offer the system the ability to learn a correction of $P$ in the natural setting. Our choice is motivated by the desire to both maintain the semantics and interpretability of $P$ and to make this processor a bottleneck forcing the model to rely on it. We show empirically that our pipeline is surprisingly robust to imperfect $P$ models even with weak (linear) encoders/decoders.

It is worth noting several potential challenges that may arise while training the natural pipeline, especially if the training data for it is sparsely available. We also suggest remedies for each:

- If $\mathbf{x}$ and/or $\mathbf{y}$ exhibit any nontrivial geometry, simple linear projections will rarely suffice for $\tilde{f}$ and $\tilde{g}$. For example, our natural inputs will often be pixel-based, necessitating a convolutional neural network for $\tilde{f}$.

- Further, since the parameters of $P$ are kept frozen, $\tilde{f}$ is left with a challenging task of mapping natural inputs into an appropriate manifold that $P$ can meaningfully operate over. While we demonstrate clear empirical evidence that such meaningful mappings definitely occur, we remark that its success may hinge on carefully tuning the hyperparameters of $\tilde{f}$.

- A very common setting assumes that the abstract inputs and latents are factorised into objects, but the natural inputs are not. In this case, $\tilde{f}$ is tasked with predicting appropriate object representations from the natural inputs. This is known to be a challenging feat (Greff et al., 2020), but can be successfully performed. Sometimes arbitrarily factorising the feature maps of a CNN (Kipf et al., 2019) is sufficient, while at other times, models such as slot attention (Locatello et al., 2020) may be required.

- One corollary of using automated object extractors for $\tilde{f}$ is that it's very difficult to enforce their slot representations to line up in the same way as in the abstract inputs. This implies that $P$ should be permutation equivariant (and hence motivates using a GNN for it).

## 4 RMR FOR BOUNCING BALLS

To evaluate the capability of the RMR pipeline for transfer from the abstract space to the pixel space, we apply it on the "bouncing balls" problem. The bouncing balls problem is an instance of a physics simulation problem, where the task is to predict the next state of an environment in which multiple balls are bouncing between each other and a bounding box. Though this problem had been studied in the the context of physics simulation from (abstract) trajectories (Battaglia et al., 2016) and from (natural) videos (Watters et al., 2017; Van Steenkiste et al., 2018; Löwe et al., 2020), here we focus on the aptitude of RMR to transfer learned representations from trajectories to videos.

Our results affirm that strong abstract models can be trained on such tasks, and that including them in a video pipeline induces more robust representations. See Appendix A for more details on hyperparameters and experimental setup.

**Preliminaries**  Here, trajectories are represented by 2D coordinates of 10 balls through time, defining our abstract inputs and outputs $\bar{\mathcal{X}} = \bar{\mathcal{Y}} = \mathbb{R}^{10 \times 2}$. We slice these trajectories into a series of moving windows containing the input, $\bar{\mathbf{x}}^*$, spanning a history of three previous states, and the target, $\bar{\mathbf{y}}$, representing the next state. We obtain these trajectories from a 3D simulator (MuJoCo (Todorov et al., 2012)), together with their short-video renderings, which represent our natural input and output space $\mathcal{X} = \mathcal{Y} = \mathbb{R}^{64 \times 64 \times 3}$. Our goal is to train an RMR abstract model on trajectories and transfer learned representations to improve a dynamics model trained on these videos.

**Abstract pipeline**  So as to model the dynamics of trajectories, we closely follow the RMR desiderata for the abstract model. We set $f$ to a linear projection over the input concatenation, $P$ to a Message Passing Neural Network (MPNN), following previous work (Battaglia et al., 2016; Sanchez-Gonzalez et al., 2020), and $g$ to a linear projection.

Our model learns a transition function $g(P(f(\bar{\mathbf{x}}^*)) \approx \bar{\mathbf{y}}$, supervised using Mean Squared Error (MSE) over ball positions in the next step. It achieves an MSE of $4.59 \times 10^{-4}$, which, evaluated qualitatively, demonstrates the ability of the model to predict physically realistic behavior when unrolled for 10 steps (the model is trained on 1-step dynamics only). Next, we take the processor $P$ from the abstract pipeline and re-use it in the natural pipeline.

**Natural pipeline**  Here we evaluate whether the pre-trained RMR processor can be reused for learning the dynamics of the bouncing balls from videos. The pixel-based encoder $\tilde{f}$ is concatenation of per-input-image Slot Attention model (Locatello et al., 2020), passed through a linear layer, and a Broadcast Decoder (Watters et al., 2019) for the pixel-based decoder $\bar{g}$.

The full model is a transition function $\bar{g}(P(\bar{f}(\mathbf{x}^*)) \approx \mathbf{y}$, supervised by pixel reconstruction loss over the next step image. We compare the performance of the RMR model with a pre-trained processor $P$ against a baseline in which $P$ is trained fully end-to-end. The RMR model achieves an MSE of $\mathbf{7.94}\pm0.41$ ($\times10^{-4}$), whereas the baseline achieves $9.47\pm0.24$ ($\times10^{-4}$). We take a qualitative look at the reconstruction rollout of the RMR model in Figure 3, and expose the algorithmic bottleneck properties for this task in Appendix C.

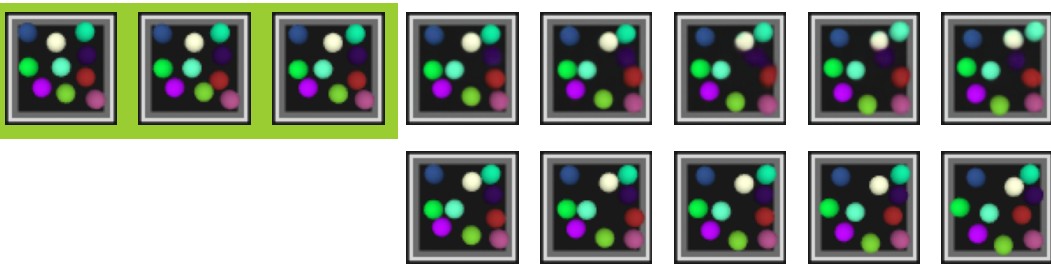

Figure 3: RMR for bouncing balls reconstruction rollout. States marked in green are the natural input, followed by the reconstructed output. The states below the reconstruction is the ground truth.

## 5 CONTRASTIVE RMR FOR ATARI

We evaluate the potential of our RMR pipeline for state representation learning on the Atari 2600 (Bellemare et al., 2013). We find the RMR applicable here because there is a potential wealth of information that can be obtained about the Atari's operation—namely, by inspecting its RAM traces.

**Preliminaries** Accordingly, we will define our set of abstract inputs and outputs as Atari RAM matrices. Given that the Atari has 128 bytes of memory, $\mathcal{X} = \mathcal{Y} = \mathbb{B}^{128\times8}$ (where $\mathbb{B} = \{0, 1\}$ is the set of bits). We collect data about how the console modifies the RAM by acting in the environment and recording the trace of RAM arrays we observe. These traces will be of the form $(\bar{\mathbf{x}}, a, \bar{\mathbf{y}})$ which signify that the agent's initial RAM state was $\bar{\mathbf{x}}$, and that after performing action $a \in \mathcal{A}$, its RAM state was updated to $\bar{\mathbf{y}}$. We assume that $a$ is encoded as an 18-way one-hot vector.

We would like to leverage any reasoning module obtained over RAM states to support representation learning from raw pixels. Accordingly, our natural inputs, $\mathcal{X}$, are pixel arrays representing the Atari's framebuffer.

Mirroring prior work, we perform contrastive learning directly in the latent space, and set $\mathcal{Y} = \mathcal{Z}$; that is, our natural outputs correspond to an estimate of the "updated" latents after taking an action. All our models use latent representations of 64 dimensions per slot, meaning $\mathcal{Z} = \mathbb{R}^{128\times64}$.

We note that it is important to generate a diverse dataset of experiences in order to train a robust RAM model. To simulate a dataset which might be gathered by human players of varying skill, we sample our data using the 32 policy heads of a pre-trained Agent57 (Badia et al., 2020). Each policy head collects data over three episodes in the studied games. Note that this implies a substantially more challenging dataset than the one reported by Anand et al. (2019), wherein data was collected by a purely random policy, which may well fail to explore many relevant regions of the games.

**Abstract pipeline** Firstly, we set out to verify that it is possible to train nontrivial Atari RAM transition models. The construction of this abstract experiment follows almost exactly the abstract RMR setup: $f$ and $g$ are appropriately sized linear projections, while $P$ needs to take into account which action was taken when updating the latents. To simplify the implementation and allow further model re-use, we consider the action a part of the $P$'s inputs. See Appendix F for detailed equations.

This implies that our transition model learns a function $g(P(f(\bar{\mathbf{x}}), a)) \approx \bar{\mathbf{y}}$. We supervise this model using binary cross-entropy to predict each bit of the resulting RAM state. Since RAM transitions are assumed deterministic, we assume a fully Markovian setup and learn 1-step dynamics.

For brevity purposes, we detail our exact hyperparameters and results per each Atari game considered in Appendix B. Our results ascertain the message passing neural network (MPNN) (Gilmer

et al., 2017) as a highly potent processor network in Atari: it ranked most potent in 17 out of 24 games considered, compared to MLPs and Deep Sets (Zaheer et al., 2017). Accordingly, we will focus on leveraging pre-trained MPNN processors for the next phase of the RMR pipeline.

**Natural pipeline**  We now set out to evaluate whether our pre-trained RMR processors can be meaningfully re-used by an encoder in a pixel-based contrastive learning pipeline.

For our pixel-based encoder $\tilde{f}$, we use the same CNN trunk as Anand et al. (2019)—however, as we require slot-level rather than flat embeddings, the final layers of our encoder are different. Namely, we apply a $1 \times 1$ convolution computing $128m$ feature maps (where $m$ is the number of feature maps per-slot). We then flatten the spatial axes, giving every slot $m \times h \times w$ features, which we finally linearly project to 64-dimensional features per-slot, aligning with our pre-trained $P$. Note that setting $m = 1$ recovers exactly the style of object detection employed by C-SWM (Kipf et al., 2019). Since our desired outputs are themselves latents, $\tilde{g}$ is a single linear projection to 64 dimensions.

Overall, our pixel-based transition model learns a function $\tilde{g}(P(\tilde{f}(\mathbf{x}), a)) \approx \tilde{f}(\mathbf{y})$, where $\mathbf{y}$ is the next state observed after applying action $a$ in state $\mathbf{x}$. To optimise it, we re-use exactly the same TransE-inspired (Bordes et al., 2013) contrastive loss that C-SWM (Kipf et al., 2019) used.

Once state representation learning concludes, all components are typically thrown away except for the encoder, $\tilde{f}$, which is used for downstream tasks. As a proxy for evaluating the quality of the encoder, we train linear classifiers on the concatenation of all slot embeddings obtained from $\tilde{f}$ to predict individual RAM bits, exactly as in the abstract model case. Note that we have *not* violated our assumption that paired $(\mathbf{x}, \bar{\mathbf{x}})$ samples will not be provided while training the natural model—in this phase, the encoder $\tilde{f}$ is frozen, and gradients can only flow into the linear probe.

Our comparisons for this experiment, evaluating our RMR pipeline against an identical architecture with an unfrozen $P$ (equivalent to C-SWM (Kipf et al., 2019)) is provided in Table 1. For each of 20 random seeds, we feed identical batches to both models, hence we can perform a paired Wilcoxon test to assess the statistical significance of any observed differences on validation episodes. The results are in line with our hypothesis: representations learnt by RMR are significantly better ($p < 0.05$) on 15 out of 24 games, significantly worse only on Pitfall!—and indistinguishable to C-SWM's on others. This is despite the fact their architectures are identical, indicating that the pre-trained abstract model induces stronger representations for predicting the underlying data factors. As was the case for bouncing balls, we expose the algorithmic bottleneck here too; see Appendix C.

As a relevant initial baseline—and to emphasise the difficulty of our task—we also include in Table 1 the performance of the latent embeddings extracted from a pre-trained Agent57 (Badia et al., 2020). These embeddings are, perhaps unsurprisingly, substantially worse than both the RMR and C-SWM models. As Agent57 was trained on a reward-maximising objective, its embeddings are likely to capture the controllable aspects of the input, while filtering out the environment's background.

**Abstract model transfer**  While the results above indicate that endowing a self-supervised Atari representation learner with knowledge of the underlying game's RAM transitions yields stronger representations, one may still argue that this constitutes a form of "privileged information". This is due to the fact we knew upfront which game we were learning the representations for, and hence could leverage the specific RAM dynamics of this game.

In the final experiment, we study a more general setting: we are given access to RAM traces showing us how the Atari console manipulates its memory in response to player input, but we *cannot* guarantee these traces came from the same game that we are performing representation learning over. Can we still effectively leverage this knowledge?

Specifically, we test *abstract model transfer* in the following way: first, we take a pre-trained abstract processor network $P$ from one game (the *"train game"*) and freeze it. Then, using this processor, we perform the aforementioned natural pipeline training and testing over frames from another game (the *"test game"*). We then evaluate whether the recovered performance improves over the C-SWM model with unfrozen weights—once again using a paired one-sided Wilcoxon test over 20 seeds to ascertain statistical significance of any differences observed. The final outcome of our experiment is hence a $24 \times 24$ matrix, indicating the quality of abstract model transfer from every game to every other game. Table 1 corresponds to the "diagonal" entries of this matrix.

Table 1: Natural modelling results for Atari 2600. Bit-level $F_1$ reported for slots with high entropy, as in Anand et al. (2019). Results assumed **significant** at $p < 0.05$ (one-sided paired Wilcoxon test).

| Game | Agent57 | C-SWM | RMR | $p$-value |
|---|---|---|---|---|
| Asteroids | $0.514_{\pm 0.001}$ | $0.582_{\pm 0.009}$ | $\mathbf{0.593}_{\pm 0.004}$ | $\mathbf{< 10^{-5}}$ |
| Battlezone | $0.351_{\pm 0.003}$ | $0.592_{\pm 0.005}$ | $0.589_{\pm 0.007}$ | $0.056$ |
| Berzerk | $0.454_{\pm 0.084}$ | $0.463_{\pm 0.053}$ | $0.470_{\pm 0.025}$ | $0.364$ |
| Bowling | $0.554_{\pm 0.004}$ | $0.944_{\pm 0.006}$ | $0.946_{\pm 0.003}$ | $0.071$ |
| Boxing | $0.558_{\pm 0.002}$ | $0.667_{\pm 0.012}$ | $0.669_{\pm 0.011}$ | $0.215$ |
| Breakout | $0.657_{\pm 0.001}$ | $0.836_{\pm 0.009}$ | $\mathbf{0.852}_{\pm 0.008}$ | $\mathbf{< 10^{-5}}$ |
| Demon Attack | $0.539_{\pm 0.004}$ | $0.653_{\pm 0.006}$ | $\mathbf{0.658}_{\pm 0.004}$ | $\mathbf{0.002}$ |
| Freeway | $0.424_{\pm 0.052}$ | $0.912_{\pm 0.025}$ | $\mathbf{0.919}_{\pm 0.035}$ | $\mathbf{0.032}$ |
| Frostbite | $0.405_{\pm 0.001}$ | $0.580_{\pm 0.025}$ | $\mathbf{0.594}_{\pm 0.016}$ | $\mathbf{0.035}$ |
| H.E.R.O. | $0.481_{\pm 0.001}$ | $0.729_{\pm 0.026}$ | $\mathbf{0.779}_{\pm 0.021}$ | $\mathbf{< 10^{-5}}$ |
| Montezuma's Revenge | $0.743_{\pm 0.003}$ | $0.824_{\pm 0.012}$ | $0.821_{\pm 0.016}$ | $0.156$ |
| Ms. Pac-Man | $0.506_{\pm 0.001}$ | $0.599_{\pm 0.004}$ | $\mathbf{0.602}_{\pm 0.006}$ | $\mathbf{0.038}$ |
| Pitfall! | $0.495_{\pm 0.003}$ | $\mathbf{0.626}_{\pm 0.015}$ | $0.603_{\pm 0.010}$ | $\mathbf{< 10^{-5}}$ |
| Pong | $0.392_{\pm 0.001}$ | $0.750_{\pm 0.016}$ | $\mathbf{0.762}_{\pm 0.010}$ | $\mathbf{0.001}$ |
| Private Eye | $0.594_{\pm 0.001}$ | $0.863_{\pm 0.010}$ | $\mathbf{0.867}_{\pm 0.008}$ | $\mathbf{0.045}$ |
| Q*Bert | $0.536_{\pm 0.010}$ | $0.588_{\pm 0.015}$ | $0.590_{\pm 0.017}$ | $0.165$ |
| River Raid | $0.686_{\pm 0.001}$ | $0.762_{\pm 0.005}$ | $\mathbf{0.764}_{\pm 0.007}$ | $\mathbf{0.032}$ |
| Seaquest | $0.472_{\pm 0.007}$ | $0.634_{\pm 0.013}$ | $\mathbf{0.653}_{\pm 0.008}$ | $\mathbf{< 10^{-5}}$ |
| Skiing | $0.599_{\pm 0.007}$ | $0.766_{\pm 0.028}$ | $0.775_{\pm 0.014}$ | $0.174$ |
| Space Invaders | $0.588_{\pm 0.002}$ | $0.719_{\pm 0.012}$ | $\mathbf{0.761}_{\pm 0.006}$ | $\mathbf{< 10^{-5}}$ |
| Tennis | $0.533_{\pm 0.008}$ | $0.724_{\pm 0.007}$ | $\mathbf{0.729}_{\pm 0.005}$ | $\mathbf{0.007}$ |
| Venture | $0.567_{\pm 0.001}$ | $0.632_{\pm 0.005}$ | $0.633_{\pm 0.004}$ | $0.392$ |
| Video Pinball | $0.375_{\pm 0.011}$ | $0.724_{\pm 0.009}$ | $\mathbf{0.745}_{\pm 0.008}$ | $\mathbf{< 10^{-5}}$ |
| Yars' Revenge | $0.608_{\pm 0.001}$ | $0.715_{\pm 0.008}$ | $\mathbf{0.751}_{\pm 0.010}$ | $\mathbf{< 10^{-5}}$ |

The results, presented in Figure 4, testify to the performance of RMR. Representations learned by RMR transfer better in $64.6\%$ of the train/test game pairs, are indistinguishable from C-SWM in $33.7\%$ of the game pairs and perform worse than C-SWM in only $1.7\%$ of game pairs. Therefore, in plentiful circumstances, the answer to our original question is *positive*: discovering a trace of Atari RAM transitions of unknown origin can often be of high significance for representation learning from Atari pixels, regardless of whether the underlying games match.

**Qualitative analysis of transfer** While we find this result interesting in and of itself, it also raises interesting follow-up questions. Is representation learning on certain games more prone to being improved just by knowing *anything* about the Atari console? Figure 4 certainly implies so: several games (such as Seaquest or Space Invaders) have "fully-green" columns, making them "universal recipients". Similarly, we may be interested about the "donor" properties of each game – to what extent are their RAM models useful across a broad range of test games?

We study both of the above questions by performing a hierarchical (complete-link) clustering of the rows and columns of the $24 \times 24$ matrix of transfer performances, to identify clusters of related donor and recipient games. Both clusterings are marked in Figure 4 (on the sides of the matrix).

The analysis reveals several well-formed clusters, from which we are able to make some preliminary observations, based on the properties of the various games. To name a few examples:

- The strongest recipients (e.g. Yars' Revenge, H.E.R.O., Seaquest, Space Invaders and Asteroids) tend to include elements of "shooting" in their gameplay.

- Conversely, the weakest recipients (River Raid, Berzerk, Private Eye, Ms. Pac-Man and Pitfall!) are all games in which movement is generally unrestricted across most of the screen, indicating a larger range of possible coordinate values to model.

- Pong, Breakout, Battlezone and Skiing cluster closely in terms of donor properties—and they are all games in which movement is restricted to one axis only.

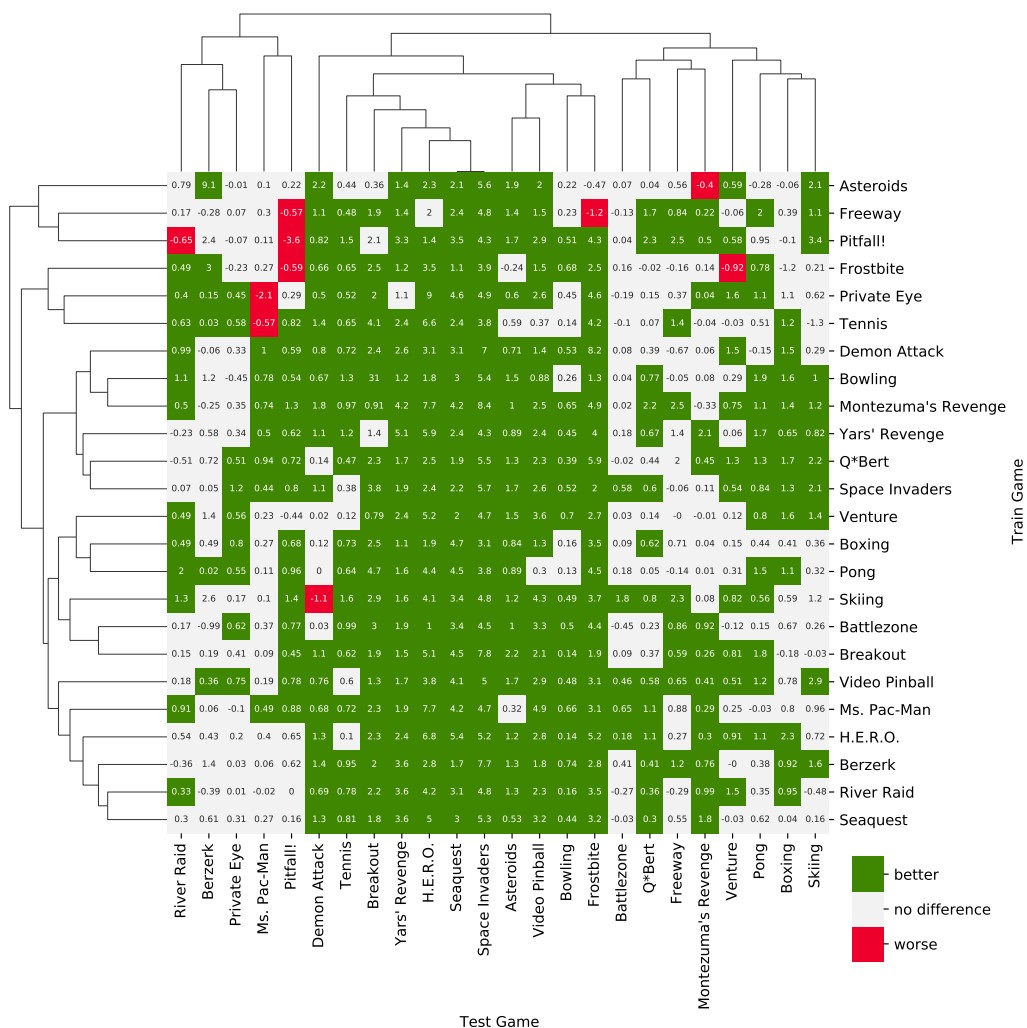

Figure 4: A hierarchically clustered heatmap depiction of the transfer results, where the *Train Game* abstract models have been trained on the *Test Game*s. The results are summarised by a Wilcoxon test, denoting where the performance of RMR is better, does not differ or is worse than C-SWM's. In each cell, the mean relative % improvement is noted. The significance cutoff is $p < 0.05$.

- Lastly, strong donors (Montezuma's Revenge, Yars' Revenge, Q*Bert and Venture) all feature massive, abrupt, changes to the game state (as they feature multiple rooms, for example). This implies that they might generally have seen a more diverse set of transitions. Further, on Yars' Revenge, a massive laser features, which, conveniently, prints an RGB projection of the RAM itself on the frame.

# 6 CONCLUSIONS

We presented Reasoning-Modulated Representations (RMR), a novel approach for leveraging background algorithmic knowledge within a representation learning pipeline. By encoding the underlying algorithmic priors as weights of a processor neural network, we alleviate requirements on alignment between abstract and natural inputs, protect our model against bottlenecks, and make it more data-efficient in the process. We believe that RMR paves the way to a novel class of algorithmic reasoning-inspired representations, with a high potential for transfer across tasks—a feat that has largely eluded deep reinforcement learning research.

## REPRODUCIBILITY STATEMENT

Our work proposes a general pipeline for representation learning under algorithmic priors. We have made a careful effort to expose all of the key elements of our RMR pipeline, both within figures and in text. We also report the hyperparameters of our experiments in the Appendices.

We carefully described the manner in which our datasets have been collected. Further, we will explore the options for open-sourcing the specific datasets we employed in this paper post-publication. We note that Atari trajectory datasets similar to the one we explore here (albeit with a significantly weaker actor) are already publicly available e.g. through the work of Anand et al. (2019).

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

## A  BOUNCING BALLS MODELLING SETUP

**Abstract pipeline**   The $f$ is a linear projection over the concatenation of inputs, outputting a 128-long representation for each of the 10 balls in the input. The $P$ is a MPNN over the fully connected graph of the ball representations. It is a 2-pass MPNN with a 3-layered ReLU-activated MLP as a message function, without the final layer activation, projecting to the same 128-dimensional space, per object. Finally, $g$ is a linear projection applied on each object representation of the output of $P$.

The model is MSE-supervised with ball position on the next step. It is trained on a 8-core TPU for 10000 epochs, with a batch size of 512, and the Adam optimizer with the initial learning rate of 0.0001.

**Natural pipeline**   $\bar{f}$ is a Slot Attention model (Locatello et al., 2020) on each input image, concatenating the images and passing them through a linear layer, outputting a 128-dimensional vector for each of the objects. $\bar{g}$ is a Broadcast Decoder (Watters et al., 2019) containing a sequence of 5 transposed convolutions and a linear layer mapping before calculating the reconstructions and their masks.

The model is MSE-supervised by pixel reconstruction (per-pixel MSE) over the next step image. Both the RMR and the baseline are trained on a 8-core TPU for 1000 epochs, with a batch size of 512, with the Adam optimiser and the initial learning rate of 0.0001, all over 3 random seeds.

## B  ATARI ABSTRACT MODELLING SETUP AND RESULTS

Our best processor network is a MPNN (Gilmer et al., 2017) over a fully connected graph (Santoro et al., 2017) of RAM slots, which concatenates the action embedding to every node (as done in Kipf et al. (2019)). It uses three-layer MLPs as message functions, with the ReLU activation applied after each hidden layer. The entire model is trained for every game in isolation, over 48 distinct episodes of Agent57 experience. We use the Adam SGD optimiser (Kingma & Ba, 2014) with a batch size of 50 and a learning rate of 0.001 across all Atari experiments. To evaluate the benefits of message passing, we also compare our model to Deep Sets (Zaheer et al., 2017), which is equivalent to our MPNN model—only it passes messages over the identity adjacency matrix. Lastly, we evaluate the benefits of factorised latents by comparing our methods against a three-layer MLP applied on the flattened RAM state.

Table 2: Abstract modelling results for Atari 2600. Entire-slot accuracies and bit-level $F_1$ scores are reported only for slots with high entropy, as per Anand et al. (2019).

| Game | Copy baseline | | MLP | | Deep Sets | | MPNN | |
|---|---|---|---|---|---|---|---|---|
| | Slot acc. | Bit $F_1$ | Slot acc. | Bit $F_1$ | Slot acc. | Bit $F_1$ | Slot acc. | Bit $F_1$ |
| Asteroids | 70.65% | 0.856 | 71.28% | 0.872 | 72.84% | 0.879 | **80.69%** | **0.930** |
| Battlezone | 57.09% | 0.841 | 61.19% | 0.840 | 61.71% | 0.867 | **71.06%** | **0.892** |
| Berzerk | 84.32% | 0.905 | 86.17% | 0.930 | 84.16% | 0.923 | **86.67%** | **0.933** |
| Bowling | 93.86% | 0.972 | 97.43% | 0.991 | 90.72% | 0.966 | **98.41%** | **0.995** |
| Boxing | 59.78% | 0.848 | 54.45% | 0.834 | **59.79%** | **0.890** | 58.56% | 0.877 |
| Breakout | 89.80% | 0.949 | 92.77% | 0.970 | 94.34% | 0.979 | **96.45%** | **0.988** |
| Demon Attack | 67.90% | 0.850 | 68.43% | 0.864 | 66.70% | 0.877 | **69.51%** | **0.879** |
| Freeway | 46.65% | 0.787 | 75.93% | 0.921 | 84.68% | 0.959 | **89.17%** | **0.965** |
| Frostbite | 76.83% | 0.904 | **79.09%** | 0.904 | 78.25% | **0.946** | 76.52% | 0.918 |
| H.E.R.O. | 76.71% | 0.891 | 82.96% | 0.932 | 80.17% | 0.929 | **89.07%** | **0.956** |
| Montezuma's Revenge | 82.58% | 0.907 | **87.30%** | 0.941 | 85.90% | **0.951** | 85.44% | 0.932 |
| Ms. Pac-Man | 83.80% | 0.941 | 80.60% | 0.935 | 81.50% | 0.952 | **85.88%** | **0.966** |
| Pitfall! | 66.60% | 0.862 | 78.28% | 0.923 | **81.92%** | **0.947** | 80.40% | 0.941 |
| Pong | 68.76% | 0.873 | 73.58% | 0.911 | 74.71% | 0.920 | **83.23%** | **0.952** |
| Private Eye | 75.25% | 0.889 | 81.95% | 0.932 | 84.77% | 0.954 | **86.41%** | **0.955** |
| Q*Bert | 83.00% | 0.915 | **90.07%** | **0.966** | 87.87% | 0.943 | 89.26% | 0.946 |
| River Raid | 76.95% | 0.895 | 80.82% | 0.927 | 69.96% | 0.865 | **86.79%** | **0.954** |
| Seaquest | 71.23% | 0.859 | **78.48%** | **0.898** | 75.53% | 0.906 | 70.94% | 0.798 |
| Skiing | 91.02% | 0.966 | 93.42% | 0.980 | 93.51% | 0.983 | **96.37%** | **0.992** |
| Space Invaders | 81.67% | 0.942 | 84.62% | 0.957 | 89.38% | 0.974 | **91.98%** | **0.985** |
| Tennis | 78.13% | 0.890 | **82.13%** | **0.926** | 71.60% | 0.856 | 80.20% | 0.893 |
| Venture | 61.29% | 0.858 | 63.16% | 0.863 | 64.88% | 0.886 | **76.56%** | **0.935** |
| Video Pinball | 76.71% | 0.848 | 85.92% | 0.912 | 78.64% | 0.877 | **86.61%** | **0.913** |
| Yars' Revenge | 69.25% | 0.896 | 74.87% | 0.929 | 72.69% | 0.948 | **84.11%** | **0.969** |

One immediate observation is that RAM updates in Atari are extremely sparse, with a *copy baseline* already being very strong for many games. To prevent the model from having to repeatedly re-learn identity functions, we also make it predict *masks* of the shape $\mathcal{M} = \mathbb{B}^{128}$, specifying which cells are to be overwritten by the model at this step. This strategy, coupled with teacher forcing (as done by Veličković et al. (2020); Strathmann et al. (2021)) yielded substantially stronger predictors. We also use this observation to prevent over-inflating our prediction scores: we only display prediction accuracy over RAM slots with label entropy larger than 0.6 (as done by Anand et al. (2019)).

The full results of training Atari RAM transition models, for the games studied in Anand et al. (2019), are provided in Table 2. We evaluate both bit-level $F_1$ scores, as well as slot-level accuracy (for which all 8 bits need to be predicted correctly in order to count), over the remaining 48 Agent57 episodes as validation. To the best of our knowledge, this is the first comprehensive feasibility study for learning Atari RAM transition models.

## C    REDISCOVERING THE ALGORITHMIC BOTTLENECK

As mentioned in the main text body, one of the key reasons in favour of a high-dimensional algorithmic component is to avoid the *algorithmic bottleneck*, as first exposed by Deac et al. (2021).

In short, the performance guarantees of running classical algorithms rely on having the exactly correct inputs for them. If there are any errors in the predictions of these (usually very low-dimensional) inputs, these errors may propagate to the algorithmic computations and yield suboptimal results. Further, there is no room for any kind of fallback if such an event occurs.

In contrast, the high-dimensional neural processors like the ones we study here are not vulnerable to bottleneck effects: if any dimensions of the latent state are poorly predicted, the other components of it could step in and compensate for this. Further, we can easily support skip connections in the case where the algorithm is not fully descriptive of the problem we're solving.

In this section, we reaffirm the bottleneck effect for both Atari and bouncing ball experiments by providing additional sets of ablations on the processor network's latent size.

We ablate various RMR processor network architectures, for $\dim \mathbf{z} \in \{2, 4, 8, 16, 32, 64\}$, but leaving all other components and operations unchanged.

The results of this ablation for the Atari representation learning setting are provided in Figure 5. It can be clearly observed that, as we reduce the size of the latents, this typically induces a performance regression in the downstream bit $F_1$ scores. This indicates the algorithmic bottleneck effect.

On the bouncing balls task, we observe the algorithmic bottleneck as well, with more pronounced effects. Namely, for all latent sizes 32 and under, the validation MSE shoots up even though the training MSE keeps improving. Ultimately, we also attempted using an Interaction Network (Battaglia et al., 2016) as a bottlenecked RMR processor which requires projecting on to $\bar{\mathbf{x}}$ rather than $\mathbf{z}$, which exhibited exactly the same behaviour, at a larger scale. The IN processor achieved a final validation MSE of $341.40 \pm 5.60 \ (\times 10^{-4})$. This is roughly $43\times$ worse than the non-bottlenecked RMR.

## D    ON THE WEAK ACCEPTOR PROPERTIES OF BATTLEZONE

Out of all the Atari games studied in our work, Battlezone could been singled out as the least "receptive" to RMR processors—with only four games successfully donating their RAM representations to its pixel based encoder (Figure 4).

We set out to study why this effect took place. Upon inspection of the game's dynamics[1], we determined that Battlezone has certain properties that make representation learning uniquely challenging and somewhat decoupled from the underlying console computation.

Namely, Battlezone is the only game in our dataset that is played in the "first-person". Therefore, from the point of view of the pixel inputs, it may seem as if the player is always in the same place, and we would expect the mechanics and the way of thinking about the 'avatar' to be substantially different from all other Atari games considered.

## E    ON THE DATA DISTRIBUTION AND SETUP USED FOR TRAINING $P$

In many cases, while an algorithmic prior may be known, generative aspects of the relevant abstract inputs for the natural task could be unknown. For example, it may be known that the natural task could benefit from a sorting subroutine, but it may not be known upfront how many objects will need to be sorted.

When such details are unknown, it may still be possible to fall back to abstract inputs sampled from some sensible generic random distribution—so long as the processor network is trained in a way that promotes extrapolation. For a recent theoretical treatment of OOD generalisation in algorithmic reasoners, we refer the reader to Xu et al. (2020). Further, as observed by Deac et al. (2021) in the case of implicit planning, even generic random abstract distributions can promote useful transfer to noisy pixel-based acting settings such as Atari.

Lastly, it is important to ensure that, in the abstract pipeline, the processor network $P$ carries the brunt of the computational burden; otherwise, the reasoning task may be partially captured in either $f$ or $g$, and not left to $P$'s weights. In our work, we generally promote such behaviour by keeping $f$ and $g$ to be only linear layers; but even in settings where this is not appropriate, we recommend taking care to not overpower the abstract encoder and decoder.

## F    ATARI ABSTRACT MODEL EQUATIONS

In this appendix, we provide a "bird's eye" view of the modelling steps taken by our abstract Atari pipeline, aiming to support future implementations of the RMR blueprint. We will readily re-use the notation from the main text for the various components.

---

[1] https://www.youtube.com/watch?v=9X4_xy7rC1A

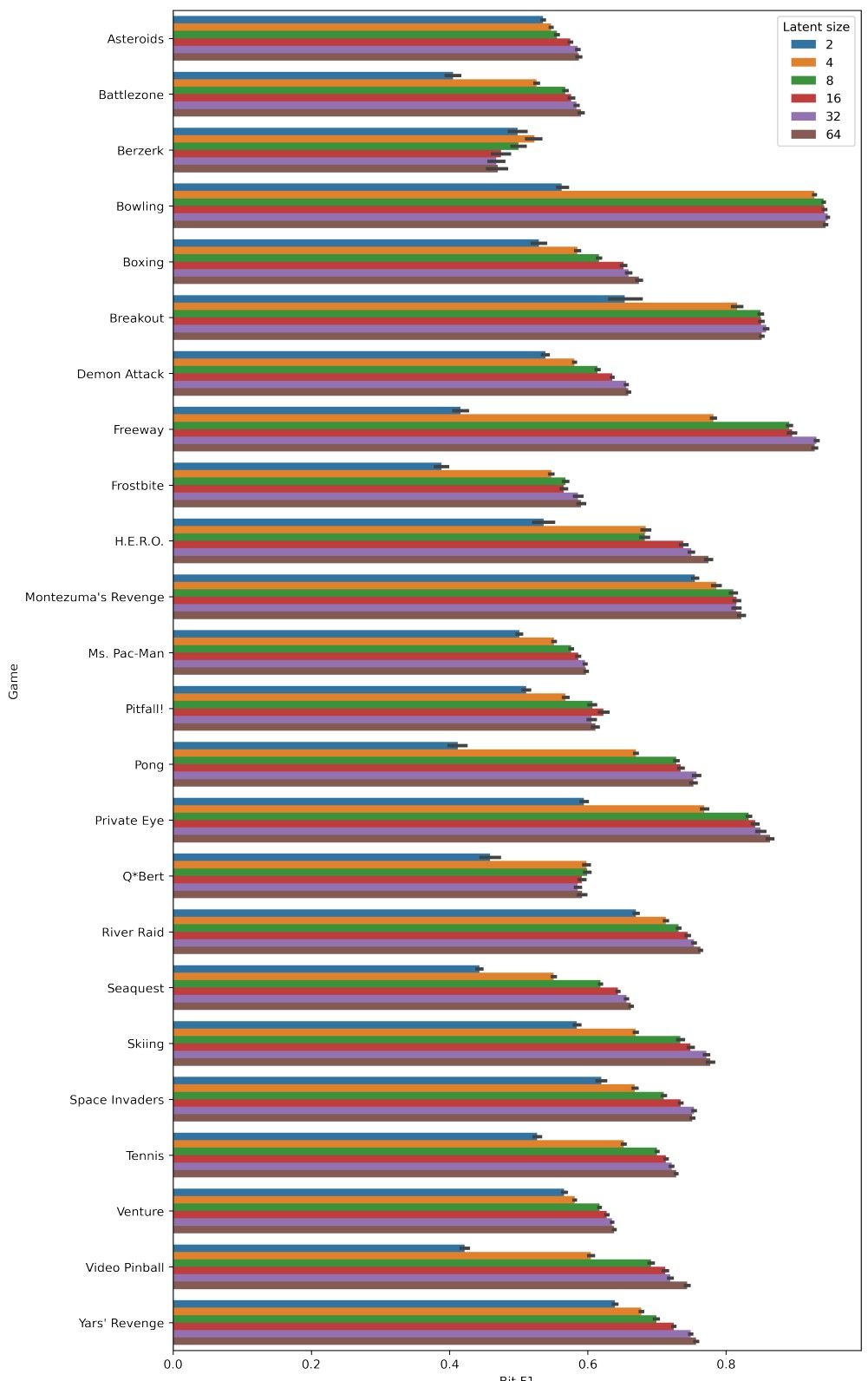

Figure 5: Bit-level $F_1$ scores for the Atari experiments while varying the latent size. A clear decreasing trend is apparent a $\dim \mathbf{z}$ is reduced, indicating the bottleneck effect.

Firstly, the abstract encoder $f$ is applied on the relevant Atari RAM representations, $\bar{\mathbf{x}}$, augmented by a one-hot representation which is aiming to provide a generic embedding of the semantics of each RAM slot. $f$ is implemented as a linear layer, hence:

$$\mathbf{z} = f(\bar{\mathbf{x}}\|\mathbf{o}) = \mathbf{W}^f \bar{\mathbf{x}} + \mathbf{U}^f \mathbf{o} + \mathbf{b}^f \tag{1}$$

where $\mathbf{W}^f, \mathbf{U}^f, \mathbf{b}^f$ are learnable weights, and $\mathbf{o} \in \mathbb{B}^{128 \times 128}$ is a one-hot encoding s.t. $\mathbf{o} = \mathbf{I}_{128}$.

In a separate pipeline (officially part of the processor network), the performed actions are also encoded using a linear action encoder:

$$\boldsymbol{\alpha} = f_a(\mathbf{a}) = \mathbf{W}^{f_a} \mathbf{a} + \mathbf{b}^{f_a} \tag{2}$$

where $\mathbf{a}$ is a one-hot encoded action representation.

Then, the processor network GNN is called to update these latents, and a sum-based skip connection is employed to promote a model-free path:

$$\mathbf{z}' = P(\mathbf{z} + \boldsymbol{\alpha}) + \mathbf{z} + \boldsymbol{\alpha} \tag{3}$$

Here, the processor network $P$ is implemented as a standard message passing neural network (Gilmer et al., 2017) over a fully connected graph. Equations of such a $P$ are commonly exposed in e.g. Veličković et al. (2019).

Lastly, the relevant decoder networks predict two properties: a mask which signifies which RAM slots have been changed as a result of applying $\mathbf{a}$, and the updated states $\bar{\mathbf{y}}$.

$$\boldsymbol{\mu} = g_\mu(\mathbf{z}') = \sigma\left(\mathbf{W}^\mu \mathbf{z}' + \mathbf{b}^\mu\right) \tag{4}$$
$$\bar{\mathbf{y}} = g(\mathbf{z}') \odot \mathbb{I}_{\boldsymbol{\mu}>0.5} = \left(\mathbf{W}^g \mathbf{z}' + \mathbf{b}^g\right) \odot \mathbb{I}_{\boldsymbol{\mu}>0.5} \tag{5}$$

Here, $\sigma$ is the logistic sigmoid activation, $\odot$ is the elementwise product, and $\mathbb{I}$ is the indicator function, which thresholds the mask. While this thresholding is non-differentiable, we note that the mask values can be directly supervised from known trajectories, and at training time, teacher forcing can be applied—slotting ground-truth masks in place of the indicator function.

