# OpenReview forum: "Reasoning-Modulated Representations"
_ICLR.cc/2022/Conference — ICLR 2022 Submitted_

### Official Review · Reviewer_o4To · 2021-10-31

**Correctness:** 4
**Technical Novelty And Significance:** 2
**Empirical Novelty And Significance:** 3
**Recommendation:** 6
**Confidence:** 4

**Main Review:**

This paper is concerned with the important problem of incorporating prior knowledge about relevant knowledge for solving a task in neural networks. The paper is well written and fairly accessible and clear. The novelty of the approach is somewhat limited, since to a large extent the idea of pretraining a processor in this way was outlined previously in Velickovic & Blundell (2021). At the same time, the implementation of this idea is by no means trivial and overall the technical contribution is sufficient in my view.

The idea of training a neural processor in the way outlined in this paper is appealing, and the possibility of using it for pixel-level prediction (eg. for bouncing balls) as well as for representation learning (eg. atari) is appealing. At the same time, I don't find that the current experiments sufficiently validate its effectiveness at these tasks, and I would like to see some improvements in this regard. In particular:

* No stronger baseline is provided for the bouncing balls experiment, which leaves it unclear how well RMR performs at this task to alternative methods that also incorporate prior knowledge about the the underlying state of the environment (eg. it being composed of objects, and object interacting mostly pair-wise). From Figure 3 it can be seen that the predictions are quite different after only 5 steps from comparing the position of the dark purple ball. A stronger baseline that also incorporates assumptions about the underlying state of the world is desirable. RIMs seems like a good candidate for this or perhaps VIN.

* On bouncing balls there is a clear expectation for what the processor should learn (i.e. something like how interaction networks are designed for this task) and I would like to see a comparison to using a processor that can be understood in this way. In particular, a comparison to a pre-trained IN as a processor would reveal how much knowledge about the underlying task is distilled. A key difference between using IN and RMR is the lack of an intermediary high-dimensional space on which the processor operates (as opposed to abstract inputs). While it is intuitive that this can be desirable when both the processor and the natural encoder are relatively unstructured, it would be good to ablate this design choice.

* While the results on Atari in Table 1 are promising, it is a little concerning that these were obtained by first pre-training an Agent57 to generate sufficiently diverse RAM state data. Part of the motivation for this work is to alleviate the need to densely sample data in the natural space by incorporating prior knowledge, yet arguably this is exactly what was needed to train agent57 in the first place. As such I would like to see an attempt made at performing this experiment on transitions collected in a way that does not necessitate recording billions of pixel-level transitions. Alternatively, one could compare to a stronger baseline that leverages the data used to train agent57 for representation purposes as well.

Finally some minor comments

* When generating simulation data from abstract inputs as for bouncing balls, how is it ensured that a sufficient range of values is seen during training the processor? In the environments considered, the abstract and natural data always comes from the same simulator, but what if this is not the case? I.e. some real-world phenomena are observed for which no simulator is available, and abstract data can only be provided via a separate simulator (for which ranges must be chosen). It could be useful to comment on this.

* When training the processor for the contrastive learning task, how is it ensured that the transformation of interest is performed by the processor and not the encoder or decoder (which are discarded afterward)?

* A good reference regarding the third bullet in the intro on the "challenging feat" of learning object representations is "On the Binding Problem in Artificial Neural Networks" by Greff et al. (2020), which could be helpful to include beyond the two specific methods cited there.


**Summary Of The Paper:**

This paper contributes an implementation of the neural algorithmic reasoning blueprint outlined in Velickovic & Blundell (2021). The setting that is considered is one where, for a given complex reasoning task that one is interested in solving, there is access to information about the underlying system that should be incorporated to simplify learning. For example, in the case of making predictions from videos about the future state of a physical system composed of bouncing balls bouncing balls, given access to each ball's radius, position, and velocity it is well known how their future movements follow the laws of physics.

In order to incorporate this prior knowledge, Velickovic & Blundell (2021) proposed the idea of training a neural processor to capture this prior knowledge between abstract inputs (eg. the 3-dimensional state of each ball) under the assumption that sufficient data obeying the known input-output mapping in this space can be generated (eg. pairs of states at time t and time t+1). An encoder then maps abstract inputs to high dimensional space, the processor transforms it to another high-dimensional space, and a decoder maps it back onto the corresponding abstract output pair. In this way, the processor can distill knowledge about the transformation between abstract inputs (eg. governing the laws of physics in its weights) into its weights. By incorporating this processor in a network that is trained to solve the original task (eg. prediction in pixel spaces) it is expected that this extracted prior knowledge can be leveraged to improve learning and transfer to new tasks.

The implementation considered here (Reasoning-Modulated Representations), is evaluated on two environments: the bouncing balls environment, and various Atari games. On bouncing balls it is shown how RMR can transfer learned representations from trajectories to videos. However, the absence of a stronger baseline makes it difficult to interpret these results. On Atari, RMR is trained on transitions of RAM states to improve predicting latent space transitions from pixels. Compared to C-SWM (Kipf et al., 2019), it is shown how RMR leads to better representations in several games, and it is shown how processors may transfer across games.


**Summary Of The Review:**

This paper contributes an implementation of an interesting approach to incorporating prior knowledge about a problem in neural networks that was outlined previously. While this constitutes a sufficient contribution, the experimental evaluation should be improved to validate the proposed implementation. I have suggested a number of concrete improvements that can be made, which would have me reconsider the current score.

---

> ### Author Response · Authors · 2021-11-23
> **Reply to Reviewer o4To**
>
> We would like to thank you for the careful and detailed assessment of our work!
>
> Our response is provided below, accompanying our recent paper revision. We hope that it appropriately addresses your comments.
>
> ### **Ablation studies**
>
> We directly ablate against several bottlenecked versions of the processor network, including using an IN-style processor (see Appendix C in the updated paper). In all cases, we successfully demonstrate the algorithmic bottleneck in this setting (larger latent sizes are more preferable). Specifically, for deploying the IN-style processor inside the natural pipeline, we found catastrophic overfitting effects, with validation MSE reaching 45x poorer values than the non-bottlenecked RMR.
>
> ### **On comparisons to stronger priors**
>
> While we agree that a comparison of RMR to alternative prior-incorporating methods could be useful, we also believe such comparisons are tangential to our setting. We want to design powerful methods in the presence of a minimal algorithmic prior: i.e., we have access to input/output pairs of the algorithm, but aren’t necessarily able to extract strong inductive biases from that algorithm.
> It would not be surprising if encoders or processor networks with stronger inductive biases would perform better. Though, if these modified modules are also parametric, they may also be amenable to RMR-like ideas, since RMR is effectively a method for setting some of the natural model’s parameters upfront.
>
> Specifically, RIMs are not a very suitable method for this since they learn independent dynamics modules per object, and the datasets employed are binary images. VINs are also not directly comparable to our work: they predict the abstract inputs from the natural inputs, implying existence of a paired and aligned dataset, which our setting does not assume. From an architectural point of view, VINs are already highly related to our existing baseline model: a CNN followed by an all-pairwise GNN over the CNN’s inferred slots.
>
> ### **On the Agent57 dataset**
>
> We acknowledge your concern about the data needed to pre-train Agent57. However, we stress that this was done solely for the purpose of stronger evaluation. We analysed the datasets generated by weaker agents (e.g. the random agent, as studied by Anand et al. (NeurIPS’19)) and we found the data insufficiently diverse to robustly claim which method is stronger; simply put, the random agent will not explore sufficiently many regions of the game. We could have, just as easily, have used human expert-based trajectories, or inserted randomised RAM states / actions into the emulator – a pre-trained Agent57, known to achieve superhuman performance in 57 Atari games, was simply a convenient proxy. The Agent57 transition data over frames is also available to the baseline models when doing natural training.
>
> ### **On the pre-training distribution for the processor**
>
> Thank you for your remark regarding the distribution used to train the processor. This is a relevant point and we now discuss it in Appendix E . In short, the relevant distributions can be determined based on any prior knowledge of the natural inputs to be expected. However, even when such information is unavailable, prior work has demonstrated that even purely “oblivious” random abstract data distributions can lead to useful processors. For more details, see XLVIN (Deac et al., NeurIPS’21).
>
> ### **On the contrastive learning task**
>
> Regarding the contrastive learning task, perhaps there was a minor misunderstanding: our processor is still trained to reconstruct the full RAM state dynamics. When contrastive learning is initiated, the processor is already pre-trained. However, to answer a more broader form of your question: we have typically managed to ensure that the processor carries the “brunt” of the abstract processing by making abstract encoders and decoders very lightweight modules (e.g. simple linear projections). We have now added this brief discussion in Appendix E.
>
> ### **On Greff et al.**
>
> Thank you for referring to the work of Greff et al., which we have incorporated into the citations.

---

> > ### Comment · Reviewer_o4To · 2021-11-30
> > **Reply**
> >
> > Thank you for your detailed reply and for addressing some of the issues I pointed out. The bottleneck ablation and the IN processor experiment are interesting and support the working of the method. I am somewhat surprised that the pre-trained IN processor performs this poorly, although it clearly indicates that the RMR pipeline is the superior alternative to mapping onto an IN.
> >
> > I would have still liked to see a stronger baseline considered for bouncing balls, since it would indicate how good the performance of RMR actually is. Currently, it is only shown that it improves over weak baselines, yet it remains unclear how close RMR is able to get to SOTA methods (even if they potentially incorporate stronger inductive biases). This is partially because the MSE and the empirical plots are difficult to interpret. Regarding Agent57 my initial concern remains. I understand that this was done to yield a stronger evaluation, however the scenario that Agent57 is used to mitigate, i.e. that of limited and biased data due to a lack of exploration, is arguably precisely what will be encountered in many real-world settings. Thus, although evaluating using Agent57 is fine, it is important to also demonstrate whether RMR can deliver in a more real-world setting.
> >
> > In light of this, I am willing to increase my score by 1 point to a 6, although I still find that the paper is borderline. I also agree with some of the critique from the other reviewers, especially regarding the real-world applicability of this set-up.

---

### Official Review · Reviewer_HnMg · 2021-11-01

**Correctness:** 4
**Technical Novelty And Significance:** 2
**Empirical Novelty And Significance:** 3
**Recommendation:** 5
**Confidence:** 4

**Main Review:**

**Strengths**:
\
The paper is well presented, and explores an important and exciting territory. The empirical assessment, though rudimentary, still demonstrates great potential of the proposed approach.

**Weaknesses**:
1. My major concern is that abstract inputs, $\bar{\bf{x}}$, is given, thus substantially simplifies the problem. While in most real-world problems, $\bar{\bf{x}}$ is unknown. (For example, the representation of sentences, speech, or images for object detection)
\
The key difficulty of representation learning is to learn internal representations for natural inputs (images here). However, the authors pre-define the abstract inputs based on their knowledge. In bouncing balls problem, they define $\bar{\bf{x}}$ to be coordinates of balls, and those are the key to predict future trajectories.
\
Given the pre-defined internal representations, it becomes much simpler to supervise a task. And $P$ is learned easily. It is not surprising that $P$ can be directly (re)-used in a natural setting, because the only thing left to do is to learn a mapping between abstract representations and natural representations, i.e., $\tilde{f}$ and $\tilde{g}$.
\
I think a more meaningful setting is that the form of abstract features ($\bar{\bf{x}}$ and $\bar{\bf{y}}$) is unknown. And the goal is to use some external knowledge to learn them instead of defining them for tasks.

2. I also found tasks being evaluated are still toy tasks. I’d suggest test on more realistic tasks, such as visual question answering.

3. There isn’t significant improvement compared to an end-end model (The difference might not be even distinguishable in the reconstruction rollout..); while RMR involves much more complicated modules. I’m not fully convinced to use RMR at least for bouncing balls problem.

4. Not weakness, but a question for style/typo: In Section 5 **natural pipeline**, is there a typo in the function? The right hand side of equation should be $\bf{y}$, not $\tilde{f}(\bf y)$?

**Summary Of The Paper:**

This paper presents a learning paradigm referred to as "Reasoning-Modulated Representation" (RMR). The main argument is that it first learns a latent-space processor for abstract simplification of a problem, then use the processor for any tasks that can be modeled with algorithmic reasoning. The authors then verify RMR in two scenarios, bouncing balls problem (physics simulation) and Atari 2600 games.

For real-world problems, the proposed method alleviates requirements on alignment between abstract and natural inputs, making the learning pipeline more data-efficient, because the pipeline could rely solely on automatically generated, abstract training data. In addition, it is possible that the latent-space processor can be re-used to tasks that do not even directly align with the abstract environment.

**Summary Of The Review:**

The paper tries to tackle a meaningful and intriguing problem. However, I find the pre-assumption of the whole learning paradigm is questionable. And I do not recommend acceptance of the paper in its current form.

---

> ### Author Response · Authors · 2021-11-23
> **Reply to Reviewer HnMg**
>
> We would like to thank you for carefully considering our paper. We are very happy about your interest in our exploration area and your appreciation of our work's presentation. We have now updated our paper with several additional ablation experiments (see, e.g. Appendix C), which we hope you will find insightful.
>
> While going through your review, we have uncovered a number of potential misunderstandings with respect to our work’s contributions. We hope that our response will help resolve some of them, and will improve your assessment of our work. In either case, we are very happy to discuss further – please let us know if any concerns remain.
>
> ### **When are algorithmic priors reasonable?**
>
> The key assumption of our work is the existence of “algorithmic priors”. That is, we assume the existence of a mapping ($\bar{\bf x} \rightarrow \bar{\bf y}$) that would be helpful for tackling the problem at hand. This is an extremely common setting, for basically any problem where a heuristical / simulator-based approach could be employed. As summarised by Reviewer fio5, our work “considers problems where such a simulator is readily available, and so can generate lots of abstract data cheaply”.
>
> For a simple example, when finding shortest routes in a real-world road network based on real-time traffic data, one might reasonably expect algorithms like Dijkstra or A* to be appropriate, even if we don’t have a direct mapping from the raw traffic data to weighted graphs needed to run such algorithms.
>
> Note that, even in our experiments, the $\bar{\bf x}$ inputs haven’t been defined to perfectly model the task at hand. Namely, for Atari, our processor network only models RAM transitions over many clock cycles, and does not consider other important (but not exposed) states, such as the Atari CPU’s registers, the game cartridge’s ROM, etc.
>
> ### **On (statistical) significance**
>
> Regarding the significance of our improvements, we would like to highlight that all our experimental results have undergone statistical significance tests, over 20 random seeds for each Atari experiment. We, indeed, demonstrate statistical significance in many of our outperformance results.
>
> ### **On RMR's complexity w.r.t. the baseline**
>
> Further, your review makes a point about RMR using “much more complicated modules … than an end-to-end model”. RMR and the respective baseline model **are identical in architecture**. RMR does not include any additional modules – it merely freezes the weights in one part of the end-to-end model (the processor, $P$) to capture the algorithmic prior.
>
> ### **On the Equation in Section 5**
>
> Lastly, regarding your note on the right-hand side of an Equation in Section 5, we would like to confirm that the RHS should be $\tilde{f}({\bf y})$. This is because our natural encoders are not trained to reconstruct raw Atari frames; rather, we perform contrastive learning based on latent-space representations of frames, as given by $\tilde{f}$. It is well known that this is a preferred approach for Atari state representation learning; for more detailed results, see Anand et al. (NeurIPS’19).

---

> > ### Comment · Reviewer_HnMg · 2021-11-29
> > **Thanks for the clarification! Some of my concerns still remain.**
> >
> > Thanks to the authors for the thorough reply. And I appreciate the authors' efforts to improve the paper.
> >
> > **Algorithmic priors**: I'm still not fully convinced that the assumption of "algorithmic priors exist" can apply widely. It is possible that the simulator can generate synthetic data cheaply, though. The application of RMR to real-world problems is restricted.
> >
> > **Empirical results**: My concerns in 2 & 3 still remain. That is, the tasks being toy, and there isn’t much improvement compared to an end-end model.
> >
> > **Significance & RMR's complexity & Equation in Section 5**: Thanks for the clarification!

---

### Official Review · Reviewer_Yu3T · 2021-11-02

**Correctness:** 3
**Technical Novelty And Significance:** 2
**Empirical Novelty And Significance:** 2
**Recommendation:** 5
**Confidence:** 4

**Main Review:**

I think this approach could potentially be useful, but I think the current set of experiments don't adequately demonstrate that it poses much of an advantage or will be scalable in a practical manner, for a number of reasons:

- The central assumption of the proposed approach is that we have access to a groundtruth algorithm that governs the interaction of entities in the target domain. This works well in the synthetic domains that are evaluated in this paper, but it's unclear to me whether this approach will be useful in real-world settings. One possibility is that perhaps a simulation engine could be made realistic enough to permit meaningful transfer to real-world data. Is that the sort of approach that the authors envision? If so, it would be helpful to include some kind of proof-of-principle demonstration for this strategy.
- Given that one has access to such a groundtruth algorithm, wouldn't it be more straightforward to simply use supervised learning to learn representations of the naturalistic inputs of the form that the algorithm expects, and then directly use the algorithm, rather than a learned proxy for it? The paper states that this approach is infeasible because it would require a 'massive dataset of paired $(x,\bar{x})$' for the purposes of training the encoder (and a similar dataset for training the decoder). But why should this dataset need to be any more massive than the one that is necessary to train the encoder/decoder in the context of the learned interaction model? The paper also suggests that this approach may be impractical because of imperfections in the groundtruth algorithm (or the information available to it). Both of these points would be far more convincing if empirically evaluated (e.g. by demonstrating that it requires more training to learn an effective encoder directly via supervised learning vs. using the current approach), and this would help to justify the motivation for the proposed approach.
- The primary contributions emphasized in the conclusion are that the proposed approach protects against bottlenecks, and that it is more data-efficient, but as far as I can tell there is no systematic evaluation of either of these claims. The claim about protection against bottlenecks could be evaluated by systematically manipulating the size of the latent representations $z$ (smaller $z$ should impair performance). The claim about data efficiency could be evaluated by manipulating the size of the training set (the advantage of the proposed approach over the end-to-end model should be particularly pronounced with smaller training sets).
- Most of the comparisons in the paper are between the proposed approach and an end-to-end analog of that approach. This is an important comparison, but the authors should also clarify how these results compare to the current state-of-the-art for the metrics that are evaluated. One exception is the comparison with embeddings from Agent57, but, as the authors note, it's unsurprising that these embeddings don't work very well given that they were learned for the purpose of performing a different task, so that particularly baseline seems a bit contrived.

Other considerations:
- It's not very easy to resolve what's going on in Figure 3. It requires a very close inspection to even detect that the balls are moving, and it isn't easy to compare the predictions to the groundtruth. It might be easier to use an example with fewer balls, or to skip frames to make the movement more salient. If possible, a link to an animation might be even better.
- Can Figure 4 be made quantitative rather than qualitative, e.g. by representing the performance of RMR as a percentage of the baseline performance? It would be nice to see how much of an improvement is yielded for each pair of games.
- The qualitative clustering analyses are kind of cool, but what do we learn about the proposed approach from these analyses? These results seem to pertain primarily to the games themselves, in which case this analysis might be better suited to the supplementary material.

**Summary Of The Paper:**

This paper proposes a method for using domain knowledge to learn an interaction model given a groundtruth algorithm that operates over abstract inputs, which can then be used to learn an encoder and decoder for interfacing between this interaction model and naturalistic inputs. The method is evaluated, with favorable results, on both the bouncing balls domain and Atari suite.

**Summary Of The Review:**

The proposed approach is favorably evaluated in synthetic domains, but it is unclear from the present set of experiments whether and how this approach will be scalable to more real-world problems. There are also some additional experiments that need to be performed to evaluate some of the paper's primary claims.

---

> ### Author Response · Authors · 2021-11-23
> **Reply to Reviewer Yu3T**
>
> Thank you for the thorough and careful assessment of our work.
>
> Our response follows, accompanying our recent paper revision – we hope that it appropriately addresses your concerns!
>
> ### **Ablation studies**
>
> Thank you for your suggestions on further ablations for the bottleneck effect. We now study the effect of the bottleneck by varying the dimensionality of $\bf z$ in our experiments (see Appendix C in the updated paper). Our results clearly indicate the algorithmic bottleneck in both Atari and bouncing balls.
>
> Further, we attempted to use a relevant (Interaction Network-style) bottlenecked processor, where the encoder has to predict abstract inputs directly. In this case, we found catastrophic overfitting effects, with validation MSE reaching 45x poorer values than the non-bottlenecked RMR.
>
> ### **Relative improvements for Atari transfer**
>
> Thank you for your suggestions regarding Figure 4. We now provide the relative % improvements w.r.t. C-SWM in every cell of this figure. Further, we agree that the clustering analyses are primarily focused on the games’ properties, but we find such transferability studies in Atari quite rare. Therefore, we have opted to keep the clustering plot in the main paper for now.
>
> ### **On the applicability of imprecise algorithms**
>
> We do not require the algorithm to strictly influence the observed interactions or the reasoning required; in fact, our usage of the skip connection (allowing for a “algorithm-free path”) further attests to this. For the specific case of Atari we studied, the algorithm is not perfectly descriptive either. We only modelled RAM transitions over many clock cycles, and do not consider other important (but not exposed) state, such as the Atari CPU’s registers, the game cartridge’s ROM, etc.
>
> ### **On the existence of paired $({\bf x}, \bar{\bf x})$ datasets**
>
> With respect to the paired dataset of $({\bf x}, \bar{\bf x})$ entries, the issue is not the dataset’s size w.r.t. $({\bf x}, {\bf y})$, but the fact that, usually, generative factors such as $\bar{\bf x}$ are not trivially obtainable for most natural problems of interest (e.g. when observing pixel arrays of an arbitrary video game console, it would usually be very difficult to make claims about that console’s internal memory state). Hence we argue that it’s hard to obtain a paired $({\bf x}, \bar{\bf x})$ dataset for most problems of interest—of any scale—and hence we do not assume such a dataset exists in our work.
>
> ### **On our baseline vs. state-of-the-art**
>
> Regarding the state-of-the-art performance levels, we would like to note that the encoders and decoders we use already mirror recent state-of-the-art practices; for example, our bouncing ball models use the recently proposed Slot Attention encoder (Locatello et al., NeurIPS’20) and the Spatial Broadcast decoder (Watters et al., 2019). Similarly, our C-SWM baseline (Kipf et al., ICLR’20) has been recently used for Atari representation learning, with a CNN backbone exactly as employed in the state-of-the-art work of Anand et al. (NeurIPS’19).

---

> > ### Comment · Reviewer_Yu3T · 2021-11-26
> > **Reply**
> >
> > Thanks to the authors for this reply. Here are my thoughts:
> >
> > **Bottleneck effects**: Thank you for including these additional experiments, they provide some nice empirical support for the 'high-dimensional' aspect of the proposed method. Do the authors intend to perform experiments to support the claim, in the conclusion, that the proposed method is more data-efficient than the end-to-end approach?
> >
> > **Plot showing relative improvements on Atari**: Thank you for adding the quantitative improvements.
> >
> > **Baselines**: Thank you for this clarification.
> >
> > I am still unconvinced about the overall motivation and likelihood of the proposed approach being useful in a real-world setting. The authors note that it is unlikely, in a real-world setting, to be able to obtain a paired dataset of $(\textbf{x},\bar{\textbf{x}})$. This is a very reasonable claim, however my question is whether, in such settings, it will be any easier to obtain a groundtruth algorithm operating over such latent variables? In other words, are there real-world settings in which we have access to a groundtruth algorithm, or suitable proxy, which can be used to train the reasoning component of the model, even though we do not have access to a dataset of paired natural and latent representations? If so, the paper, and general approach, would be much more compelling if an empirical demonstration of such a scenario could be provided.

---

> > > ### Author Response · Authors · 2021-11-30
> > > **Further discussion**
> > >
> > > Thank you very much for acknowledging our rebuttal, and we are very glad that our additional results have made our contribution stronger.
> > >
> > > We are delighted to keep the discussion going regarding the algorithmic prior. In particular, we argue that the setting you describe:
> > > >  In other words, are there real-world settings in which we have access to a groundtruth algorithm, or suitable proxy, which can be used to train the reasoning component of the model, even though we do not have access to a dataset of paired natural and latent representations?
> > >
> > > is in fact very common.
> > >
> > > Classical algorithms are typically designed as an abstractified way of solving real-world problems. Casting real-world data into the input space of the algorithm, however, may be very tricky.
> > >
> > > One standard example is finding shortest routes in a real-world road network based on real-time traffic data. It can be argued that algorithms such as Dijkstra or A* have been invented precisely to tackle problems like these. However, the raw real-world road network data may be highly noisy, change dynamically, and even if we are able to isolate "nodes" for a relevant graph structure, there is still not a highly obvious mapping that converts the raw data into a single scalar "length" per edge.
> > >
> > > In our work, we have focused on demonstrating that, as a proof-of-concept, such algorithms can be carried meaningfully into noisy problems, and yield useful improvements to representation learning.
> > >
> > > We also intend to demonstrate data efficiency gains more explicitly. The bottleneck study already required about ~1,500 individual experiment runs, and we found it to be relevant to one of our central hypotheses, so it occupied most of our rebuttal time. But we are happy to follow up on this point too.

---

> > > > ### Comment · Reviewer_Yu3T · 2021-11-30
> > > > **Reply to discussion**
> > > >
> > > > Thanks to the authors for this additional discussion. I just want to clarify that I don't find this suggestion (that RMR can be used to connect real-world data to classical, abstract algorithms) entirely implausible. It's just that, given the present experiments, this idea is merely speculative. The present experiments only study cases in which the underlying generative model is known and can be used to train the reasoning model, which is a good starting point, but leaves open the question of whether the work can be extended to the kind of real-world cases that you suggest (e.g. the road network example). Given that, I think the work is still in 'slightly below threshold' category, but I wish the authors well in their continued efforts on this project.

---

### Official Review · Reviewer_fio5 · 2021-11-03

**Correctness:** 4
**Technical Novelty And Significance:** 2
**Empirical Novelty And Significance:** 2
**Recommendation:** 6
**Confidence:** 3

**Main Review:**

First, this paper addresses a very important topic. The fusion of deep learning and classical algorithms is a frontier in AI research with significant potential going forwards.

This paper is a somewhat unusual submission. The methodological contribution is minimal (the approach is easily described in words). The work doesn’t follow the usual sorts of patterns e.g., explaining why something works, or shedding new light on some occurrence. Instead, the main contribution of this work is to spell out a philosophy on how to combine deep learning and classical algorithms. It is for this reason that I wouldn’t be surprised if the paper receives some negative reviews. However, I myself find the conceptual contribution to be valuable, with the potential to change the way others think.

With this in mind, here a few high-level strengths and weaknesses of this work:

**Strengths:**
- The paper addresses a fundamental conceptual question, a nice change from many papers I read.
- The approach is simple and general, and could be replicated for other related problems.
- Good empirical performance compared to C-SWM.
- Excellent writing.
- Proper treatment of statistical significance. All too uncommon in our community.

**Weaknesses:**
- Light on ablations. There is a study of different architectures for $P$, but nothing beyond that. I’ll try to offer some suggestions below, but the authors probably can think of many more.
- As best I can tell the authors don’t plan on releasing code (please correct me if this is untrue).
- I wouldn’t call this work reproducible in its current state. E.g., the paper mentions in passing a skip connection in $P$ but I was unable to find further information on this (please point to it if I missed it). Hyperparameters are given, but a lot of the implementation details are left the to imagination.

**Questions and suggestions:**

- One of the central ideas in this work (also present in a couple of prior works) is that of a dimensionality bottleneck if one were to fix the reasoning module $P$ to have the same low dimensionality as the abstract inputs (size, position etc.). While this is an intuitive point, and somewhat observed already, it is not yet a widely known thing. So I would suggest an additional ablation that shows that RMR performs favorably compared to an architecture with a dimensionality bottleneck. I fully expect RMR to do better, but I would be interested in seeing the delta so I can understand how much of the work is being done by the avoidance of a dimensionality bottleneck.
- Any thoughts on why with Battlezone as the test game the performance is often “no different” from C-SWM (Fig. 4)? Perhaps the reasoning module plays a less important role for this task?
- The fact that $P$ can be transferred effectively between tasks is somewhat surprising to me. It makes me wonder: what is $P$ learning that is helping performance? It seems it isn’t the precise steps of the algorithm in question…
- The authors mention a greater need for careful hyperparameter tuning. It would be nice to quantify/demonstrate that point somehow.
- The method doesn't require $(x, \bar{x})$ pairs etc., as you mention. I am curious how close RMR comes to matching the performance of a model trained with this sort of privileged information.
- One take home from your results is that training $P$ is best done with lower dimensional abstract representations than the high dimensional raw input. It would be interesting to see some further experimentation on the relation between the dimensionality of the data $P$ is trained on, and its performance when inserted into the final model.

**Summary Of The Paper:**

This paper proposes a training method based on ideas on neural algorithmic reasoning. As well using a model architecture that is aligned with the underlying algorithmic process this method proposes to pre-train the reasoning module using data from an abstract output stimulator (the work considers problems where this simulator is readily available, and so can generate lots of abstract data cheaply). A key motivation is that the reasoning module $P$ should ultimately be a mapping between two medium dimensional latent spaces (lower dimension than input space, higher dimension than the abstract representation space). Finding this balance allows the module to avoid modeling high dimensional data, whilst also not introducing low-dimensional bottlenecks into the architecture.  This work finds that inserting the pre-trained $P$ into a model that processes raw input data leads to improved performance compared to end-to-end training of a model on raw input data.

**Summary Of The Review:**

In all, I opt for arguing for acceptance. Since the contribution of this work is somewhat subtle, I am especially open to further discussions. The basis for my current position is this:

- This works adds further weight behind a school of thought on combining deep learning and classical algorithms: instead of exactly running classical algorithms as subroutines in network architectures, we should instead insert network modules that behave similarly to classical algorithms, without exact execution.
- The key requirement in order to use this method, that there are problems where a cheap abstract simulator is available, is fairly reasonable. For example, any case in which we wish to insert a classical algorithm as a subroutine in a network architecture, we have available a simulator (i.e., the classical algorithm).
- Given the previous two points, the takeaways from this work have to potential to be of use in various problem settings.

I opt for a weak accept instead of higher since although I enjoyed reading this paper, I did feel that 1) this paper appears to be somewhat modest extension of existing ideas, and 2) I feel the paper is missing some ablations.

---

> ### Author Response · Authors · 2021-11-23
> **Reply to Reviewer fio5**
>
> We would like to thank you for your careful review of our work, and for kindly pointing out its strengths as well as suggesting relevant improvements.
>
> Our response is provided below, accompanying our recent paper revision. We hope that it appropriately addresses your comments.
>
> ### **Ablation studies**
>
> We provide two follow-up ablations (spanning ~1,500 individual model experiments) in response to your suggestions, which we found would make a strong impact on our contributions. Namely:
>
> - We study the effect of the bottleneck by varying the dimensionality of $\bf z$ in our experiments (see Appendix C in the updated paper). Our results clearly indicate the algorithmic bottleneck in both Atari and bouncing balls.
> - Further, we attempted to use a relevant (Interaction Network-style) bottlenecked processor, where the encoder has to predict abstract inputs directly. In this case, we found catastrophic overfitting effects, with validation MSE reaching 45x poorer values than the non-bottlenecked RMR.
>
> ### **Reproducibility**
>
> It is our intention to release both the Agent57 dataset used to train our Atari representation learner, and the source code of our training subroutines. We will strive to do this as soon as possible upon acceptance.
>
> We have now added Appendix F to make our architectural dataflow for the abstract Atari tasks more explicit and in equation form. We hope that this will address your concern about reproducibility.
>
> ### **On the recipient properties of Battlezone**
>
> Thank you for raising the point about Battlezone’s peculiar recipient properties – we originally missed this phenomenon. We have now inspected the game’s dynamics (https://www.youtube.com/watch?v=9X4_xy7rC1A), as well as observing the available RAM annotations for this game, as extracted by Anand et al (https://github.com/mila-iqia/atari-representation-learning/blob/master/atariari/benchmark/ram_annotations.py). We determined that it has some properties that make representation learning uniquely challenging and somewhat decoupled from the underlying console computation. Namely, Battlezone is the only game in our dataset that is played in the “first-person”. Therefore, from the point of view of the pixel inputs, it may seem as if the player is always in the same place, and we would expect the mechanics and the way of thinking about the 'avatar' to be substantially different from all other Atari games considered. We have now made this discussion clear in Appendix D.
>
> ### **Does $P$ learn precise steps?**
>
> Following up on this point, we would like to emphasise that $P$ is unlikely to learn precise steps of the Atari console’s computations. However, even at an approximate level, there is benefit to learning the kinds of processing steps that the console might be doing. On Atari, the programmers had access to only 128 bytes of memory. This implies that, very often, exceptionally clever (and reusable) strategies needed to be employed in order to make full use of this storage. We believe that, within the weights of $P$, we can capture such strategies, without complete dependence on which game is being played.

---

> > ### Comment · Reviewer_fio5 · 2021-11-29
> > **Thanks for the additional details**
> >
> > Dear authors,
> >
> > I just wanted to register my thanks for the feedback you gave based on my original comments. First it is great to have confirmation on the release of source code. Second, I'm grateful for the couple of extra perspectives you offered on the bottleneck effect. I appreciate the effort put into running these additional experiments, and even though the results are as expected still think they offer helpful clarifications.
> >
> > I will maintain my generally positive outlook on this work :)
> >
> > Best wishes

---

### Decision · Program_Chairs · 2022-01-20

**Decision:**

Reject

**Comment:**

The paper proposes Reasoning-Modulated Representations (RMR). That is, it incorporates how to incorporate (structure) prior knowledge (such as a law in physics) into a pre-trained reasoning modules, and investigates how doing so shapes the discovered representations in a number of self-supervised learning settings from pixels. The reviews and (short) discussion have presented salient arguments about the suitability of the paper for publication at this stage. One review argues that the "methodological contribution is minimal," another one is asking for "systematic evaluation" of the main claims made. Moreover, while we all agree that the direction is interesting, the RMR approach presented is not shown to "scale well" (yet), as pointed out by one review. This, however, is important since the general idea that prior knowledge shapes the representation learned is common wisdom in the literature. Indeed, one may now argue that the paper is much more about "how best to combine pixel-based deep learning and neural algorithmic reasoning algorithms" as one reviewer puts it. From this perspective the ATARI experiments are more interesting but here the benefit compared to C-SWM seems to be marginal and one should compare to other deep baseline conditions on the RAM; the significance is not looking at the difference in score and degree in freedoms but just the number of wins. Additionally, there should be other baselines that directly make use of more structured models (structure = prior knowledge, e.g., HMMs or some other way to have bit of a memory), other datasets (where no access to RAM exists) as well as a discussion of other approaches that combines (combinatorial) reasoning with pixel-based deep learning. That is, while pushing for a more high-level contributions is fine, this also requires some more illustrations and discussion of the broader context. Therefore, my overall recommendation is reject at this stage of the paper.